# Integrated Proteomics and Metabolomics Analysis of Nitrogen System Regulation on Soybean Plant Nodulation and Nitrogen Fixation

**DOI:** 10.3390/ijms23052545

**Published:** 2022-02-25

**Authors:** Xiaochen Lyu, Chunyan Sun, Jin Zhang, Chang Wang, Shuhong Zhao, Chunmei Ma, Sha Li, Hongyu Li, Zhenping Gong, Chao Yan

**Affiliations:** College of Agriculture, Northeast Agricultural University, Harbin 150030, China; xiaochenlyu@163.com (X.L.); s17860745185@163.com (C.S.); 15263485352@163.com (J.Z.); 18249555621@163.com (C.W.); shhzh091@sina.com (S.Z.); chunmm@neau.edu.cn (C.M.); risa_shasha@163.com (S.L.); lihongyu89103@126.com (H.L.)

**Keywords:** nodulation, nitrogen fixation, proteomics, metabolomics

## Abstract

The specific mechanisms by which nitrogen affects nodulation and nitrogen fixation in leguminous crops are still unclear. To study the relationship between nitrogen, nodulation and nitrogen fixation in soybeans, dual-root soybean plants with unilateral nodulation were prepared by grafting. At the third trifoliate leaf (V3) to fourth trifoliate leaf (V4) growth stages (for 5 days), nitrogen nutrient solution was added to the non-nodulated side, while nitrogen-free nutrient solution was added to the nodulated side. The experiment was designed to study the effects of exogenous nitrogen on proteins and metabolites in root nodules and provide a theoretical reference for analyzing the physiological mechanisms of the interaction between nitrogen application and nitrogen fixation in soybean root nodules. Compared with no nitrogen treatment, exogenous nitrogen regulated the metabolic pathways of starch and sucrose metabolism, organic acid metabolism, nitrogen metabolism, and amino acid metabolism, among others. Additionally, exogenous nitrogen promoted the synthesis of signaling molecules, including putrescine, nitric oxide, and asparagine in root nodules, and inhibited the transformation of sucrose to malic acid; consequently, the rhizobia lacked energy for nitrogen fixation. In addition, exogenous nitrogen reduced cell wall synthesis in the root nodules, thus inhibiting root nodule growth and nitrogen fixation.

## 1. Introduction

High soybean yields depend on the combined effects of nitrogen fixation in the root nodules and fertilization with nitrogen. However, exposing soybean roots to high concentrations of nitrogen can inhibit root nodule growth, accelerate root nodule senescence, and reduce the efficiency of nitrogen fixation [1,2]. The effects of exogenous nitrogen on nodulation and nitrogen fixation in soybeans are influenced by nitrogen concentration, application site, and treatment time, among other factors [3,4,5,6,7,8]. Exogenous nitrogen was separately supplied to the upper and lower layers of soybean roots [9,10] or supplied to the left and right sides of dual-root soybeans [11,12], and all showed a rapid and reversible effect of nitrogen on root nodule growth and nitrogenase activity. When nitrogen was directly or indirectly supplied to soybean roots, local and systemic responses of nodulation and nitrogen fixation occurred in the roots [9,10,13,14]. Tanaka et al. [15] and Xia et al. [10] conducted split-root or dual-root experiments and found that root nodules in direct contact with nitrate and ammonium were inhibited, while root nodules on the side without a nitrogen supply were not clearly inhibited. Additionally, Lyu et al. [16] applied nutrient solutions with different nitrogen concentrations to non-nodulated roots in dual-root soybean plants with unilateral nodulation, and the nodule dry weight, nodule number, and nitrogenase activity on the side without the nitrogen supply decreased significantly with increasing concentrations of nitrogen, thus supporting systemic regulation.

Experiments using ^13^C and ^14^C labeling showed that exogenous nitrogen reduced nitrogenase activity and ^13^C and ^14^C abundance in soybean root nodules, and the nitrogen-mediated inhibition of nodulation and nitrogen fixation was associated with a decrease in photosynthate distribution in root nodules [7,17]. Some scholars believe that exogenous nitrate, which enters plants first, is reduced to nitrite, and then nitrite is metabolized into NO [18,19]. The nitrogenase activity of root nodules is reduced with increasing NO concentrations in the roots [20]. Increasing the asparagine and ureide contents of the nutrient solutions of hydroponic soybeans increased their content in root nodules, and the nitrogenase activity of the root nodules was inhibited. Thus, asparagine and ureide may be crucial feedback substances that inhibit nitrogen fixation [21,22].

Changes in nitrogen fixation in root nodules are largely regulated by changes in enzyme activity, while changes in metabolites are a direct response to the regulatory processes that occur in the interaction between nitrogen application and nitrogen fixation in root nodules. In this study, dual-root soybean plants with unilateral nodulation received high concentrations of nitrogen on the non-nodulated root, and the metabolic pathways associated with the systemic regulation of nitrogen on nodulation and nitrogen fixation were studied through the combined proteomic and metabolomic analysis of the root nodules on the side without exogenous nitrogen. Theoretical references are provided to analyze the physiological mechanism of the interaction between nitrogen application and fixation in soybean root nodules.

## 2. Results

Dual-root soybean plants with unilateral nodulation were treated with a nitrogen supply at the V3 to V4 stage. Nitrogen-free nutrient solution was added to the nodulation side of all treatments. A nutrient solution with a nitrogen concentration of 100 mg/L was added to the non-nodulated roots in the NH treatment. Nitrogen-free nutrient solution was added to the non-nodulated roots in the NF treatment.

### 2.1. Effects of Nitrogen Supply on Nodule Growth and Nitrogenase Activity

Table 1 demonstrates the effects of nitrogen supply on nodule numbers and dry weight. With nitrogen supplied to non-nodulated side roots for 5 days, no significant difference in the root nodule number and weight was found on nodulated side roots. The results showed that nitrogen supplied for 5 days had no significant effect on nodule growth and development. Figure 1 shows the effects of nitrogen supply on nitrogenase activity in dual-root soybean nodules. After nitrogen was supplied to the non-nodulated side roots, compared with those with NF treatment, the specific nitrogenase activity (SNA) and acetylene reduction activity (ARA) on the nodulated side of NH treatment decreased by 44.0% and 41.0%, respectively, indicating that 5 days of nitrogen supply to non-nodulated side roots had significant effects on the ARA and SNA of nodulated side root nodules.

### 2.2. Proteomics Analysis of Soybean Nodules

An integrated approach involving high-performance liquid chromatography–tandem mass spectrometry (HPLC-MS/MS) and tandem mass tag (TMT) labeling was used to analyze the proteomics changes between the nitrogen supply treatment and the control. A total of 8573 proteins were identified, 5279 of which belonged to the Glycine max (Soybean) (*Glycine hispida*) UniProt database (Appendix A), and 3294 proteins belonged to the *Bradyrhizobium japonicum* UniProt database (Appendix A). The differentially expressed proteins (DEPs) between the treatment groups were determined by using a fold change (FC) greater than 1.2 for protein upregulation or smaller than 0.83 for protein downregulation and a *p* value < 0.05 in Student’s *t* test as criteria. A total of 164 DEPs were identified (Table 2).

First, using basic local alignment search tool (BLAST) for sequence alignment and Gene Ontology (GO) for annotation, the functions of the DEPs identified above were investigated. Based on functional features, the DEPs belonged to three groups associated with cellular components (CC), molecular functions (MF), and biological processes (BP).

The DEPs of soybean nodules tissue in the CC group were mostly enriched in the amyloplast, endoplasmic reticulum, endomembrane system, membrane protein complex, etc. The DEPs in the MF group were mostly enriched in glycogen (starch) synthase activity, cofactor binding, heme binding, tetrapyrrole binding, UDP-glycosyl transferase activity, metal ion binding, oxidoreductase activity, structural constituent of ribosome, etc. Those in the BP group were mostly enriched in starch biosynthetic process, protein exit from endoplasmic reticulum, starch metabolic process, regulation of vesicle-mediated transport, protein folding, hydrogen peroxide metabolic process, cellular amide metabolic process, organonitrogen compound biosynthetic process, regulation of transport, membrane organization, carbohydrate derivative metabolic process, organonitrogen compound metabolic process, etc. These findings indicated that nitrogen treatment can regulate carbon metabolism and nitrogen metabolism in soybean nodule tissue, thereby affecting the nitrogen fixation ability of nodules (Figure 2).

The DEPs of soybean rhizobia in the CC group were mostly enriched in the cytoplasm, nonmembrane-bound organelles, protein-containing complexes, etc. The DEPs in the MF group were mostly enriched in nitrogenase activity, oxidoreductase activity, acting on iron-sulfur proteins as donors, RNA binding, cation-transporting ATPase activity, active ion transmembrane transporter activity, ATPase coupled ion transmembrane transporter activity, etc. Those in the BP group were mostly enriched in nitrogen fixation, nitrogen cycle metabolic process, organonitrogen compound biosynthetic process, cellular process, ATP hydrolysis coupled cation transmembrane, purine ribonucleoside metabolic process, nitrogen compound metabolic process, glycosyl compound metabolic process, glutamine family amino acid metabolic process, amide biosynthetic process, cellular nitrogen compound metabolic process, etc. The results were the same as those for nodule tissue, and most of the differential proteins in rhizobia were enriched in the process of nitrogen and carbon metabolism (Figure 3).

Kyoto Encyclopedia of Genes and Genomes (KEGG) pathway analysis was conducted to explore the metabolism or signaling pathways in which the identified DEPs might be involved. As shown in Figure 4, the DEPs in soybean nodule tissue were mostly involved in valine, leucine, and isoleucine biosynthesis, one-carbon pool by folate, sesquiterpenoid and triterpenoid biosynthesis, C5-branched dibasic acid metabolism, and protein processing in the endoplasmic reticulum. As shown in Figure 5, the DEPs in soybean rhizobia were mostly involved in carbon metabolism, nitrogen metabolism, carbon fixation in photosynthetic organisms, folate biosynthesis, fructose and mannose metabolism, ribosomes, etc.

### 2.3. Metabolomics Analysis of Soybean Nodules

Changes in the metabolic profiles of the soybean nodules were determined using the metabolomics method based on hydrophilic interaction chromatography ultrahigh performance liquid chromatography coupled with tandem quadrupole time-of-flight mass spectrometry (HILIC UHPLC-Q-TOF) technology. A total of 56 differentially expressed metabolites (DEMs), including 20 upregulated and 36 downregulated DEMs were identified (Appendix A).

Figure 6 presents the metabolic pathways enriched with DEMs. The DEMs were enriched mostly in metabolic pathways, biosynthesis of secondary metabolites, biosynthesis of amino acids, pentose phosphate pathway, carbon metabolism, ABC transporters, sulfur metabolism, glycine, serine and threonine metabolism, glutathione metabolism, folate biosynthesis, cyanoamino acid metabolism, cysteine and methionine metabolism, arginine biosynthesis, and arginine and proline metabolism.

### 2.4. Integrated Proteomic and Metabonomic Analysis

An integrated KEGG pathway analysis was conducted using both the DEPs and DEMs between the two treatment groups. The KEGG pathways in soybean nodule tissue are presented in Figure 7. The pathways included metabolic pathways, biosynthesis of secondary metabolites, protein processing in the endoplasmic reticulum, carbon metabolism, ABC transporters, ribosome, biosynthesis of amino acids, RNA transport, plant-pathogen interaction, biosynthesis of cofactors, starch and sucrose metabolism, glycerophospholipid metabolism, cysteine and methionine metabolism, glycine, serine, and threonine metabolism, glyoxylate and dicarboxylate metabolism, etc. The KEGG pathways in soybean rhizobia are presented in Figure 8. The metabolic pathways involved biosynthesis of secondary metabolites, microbial metabolism in diverse environments, biosynthesis of antibiotics, carbon metabolism, ABC transporters, biosynthesis of amino acids, glycerophospholipid metabolism, pentose phosphate pathway, methane metabolism, folate biosynthesis, etc.

## 3. Discussion

### 3.1. Regulation of Nitrogen Supply on Nodules Energy Metabolism

Carbon metabolism in root nodules is the main source of energy for nitrogen fixation [23]. Sucrose can be converted into starch for storage by starch synthase (StS) and be converted into UDP-glucose and fructose to provide energy for nitrogen fixation in root nodules via catalysis by sucrose synthase (SS). Forrest et al. [24] showed a negative correlation between the ability to fix nitrogen and the starch level in soybean root nodules. In this study, StS (A0A0R0HUM4, A1YZE0) in soybean root nodules was upregulated, and exogenous nitrogen increased starch synthesis but decreased nitrogenase activity in the root nodules (Appendix A, Figure 1). SS is important in nodulation and nitrogen fixation in leguminous crops. For example, in pea mutants without SS, root nodules cannot effectively promote nitrogen fixation [25]. Gordon et al. [26] discovered that the expression of the gene regulating SS synthesis in root nodules was downregulated after the application of 10 mM nitrate to soybeans for 1 day, while SS activity decreased and the sucrose concentration increased in root nodules after 3–4 days of nitrogen application. In this study, in root nodules supplied with nitrogen, SS (I1MHJ6, K7MZJ1) was downregulated, while the sucrose concentration did not change significantly (Appendix A); this observation may have resulted from the conversion of sucrose into starch due to nitrogen application (Figure 9).

UDP-glucose and fructose generated by SS can further enter metabolic pathways such as glycolysis. Oxaloacetic acid can be converted into malic acid by malate dehydrogenase [23], and malic acid is in turn transported to the symbiont via transport proteins, thereby maintaining normal respiration and efficient nitrogen fixation in bacteroids [27,28,29,30]. Tesfaye et al. [31] showed that overexpressing malate dehydrogenase in alfalfa could lead to an increase in the malic acid concentration in alfalfa root nodules. In this study, in soybeans supplied with nitrogen, malate dehydrogenase (I1KL87, I1MTU1) was downregulated, and the malic acid concentration in root nodules decreased (Appendix A), demonstrating that the inhibition of nitrogen fixation by exogenous nitrogen may result from the rhizobia’s lack of nitrogen fixation energy since the malic acid in the root nodules was reduced (Figure 9).

### 3.2. Regulation of Nitrogen Supply on the Nodules Cell Wall

The nitrogen fixation activity of root nodules in leguminous crops is closely associated with cell wall structure [32,33,34]. UDP-xylose and UDP-rhamnose, which are important components of plant cell walls, are generated from UDP-glucose by UDP-xylose synthase (UXS) and rhamnose synthase (RHM). In this study, UXS (I1L7V1, A0A0R0ERF2) and RHM (D4Q9Z5) were downregulated (Appendix A), and UDP-xylose and UDP-rhamnose were decreased (Appendix A), indicating that exogenous nitrogen inhibited the synthesis of cell wall components. Furthermore, S-adenosylmethionine, a key substance in lignin synthesis, is converted from serine [35]. In this study, in root nodules supplied with nitrogen, the serine concentration increased, but the S-adenosylmethionine concentration decreased (Appendix A), which explained why exogenous nitrogen also inhibited the synthesis of lignin, the main component of root nodule cell walls. Although there was no significant change in the number or weight of nodules treated with nitrogen for 5 days (Table 1), previous studies have shown that a long-term nitrogen supply inhibits the growth of nodules [16], indicating that with the increase in nitrogen supply treatment time, the reduction in cell wall synthesis is an important cause of the inhibition of nodule growth. However, there was no significant difference in nodule growth in this study, which may have been due to the short treatment time. Although the lignin content was changed, the process of inhibiting cell wall synthesis was only beginning, which was not reflected in the number and dry weight of nodules(Figure 9).

### 3.3. Regulation of Nitrogen Supply on Nitrogen Fixation Signaling Substances in Nodules

Some scholars believe that putrescine [36], nitric oxide (NO) [37,38,39,40,41,42], and asparagine [43,44,45] are signaling molecules that regulate nitrogen fixation in root nodules. The concentration of putrescine in the root nodules of chickpea, vetch and blackgram is positively correlated with the ability to fix nitrogen [36,46,47]. Moreover, spraying putrescine on soybean leaves reduces the number of root nodules, while spraying brassinolide increases the putrescine concentration and decreases the number of root nodules [48]. Catalytic synthesis of putrescine from arginine is performed by ornithine decarboxylase (ODC) in plants. In this study, nitrogen supply decreased the arginine concentration, while putrescine and ODC (A0A0A3Z5A3) expression increased. Based on the precursors and enzymes, exogenous nitrogen promoted the transformation of arginine to putrescine in nodules and reduced the nitrogenase activity in soybean nodules (Appendix A), which differs from results from studies on chickpea, vetch, and blackgram [36,46,47] but is similar to results from soybeans [48], suggesting that the putrescine-mediated regulation of nitrogen fixation may be related to the crop species. Maskall et al. [49] believed that NO inhibited nitrogenase activity because soybean hemoglobin has a higher affinity for NO than for O_2_ in root nodules. After Lotus was supplied with nitrate for 27 h, the NO concentration in root nodules increased, and the ARA decreased [20]. In this study, NO was not detected in the metabolomics results, but NO synthase (NOS) (A0A6G0U7K1) was upregulated in nodules (Appendix A). We speculated that the nitrogen supply promoted NO synthesis in nodules. To verify this result, we used physiological methods to determine the NO content in nodules and visualized the NO production in nodules. As expected, the nitrogen increased NO levels in nodules (Appendix A, Appendix A). Heme is the core cofactor for functional soybean hemoglobin. Nishiwaki et al. [50] found that the concentration of hemoglobin in soybean root nodules decreased slightly because of the nitrate supply. In this study, downregulation of the four heme binding proteins (I1L0D9, I1MBJ1, O48560, and Q9XHC6) in root nodules supplied with nitrogen revealed that hemoglobin synthesis in soybean root nodules was inhibited by exogenous nitrogen (Appendix A), and changes in the soybean hemoglobin content might be regulated by NO. Lyu et al. [16] supplied nitrogen to soybeans during the V4-R3 stages and found no obvious effect on SNA at the R3 stage, and the reduction in root nodules reduced the ARA. In addition, Li et al. [51] showed that supplying nitrogen to soybeans for 3 days at the R1 stage inhibited ARA. In this study, SNA significantly decreased during nitrogen treatment, and three proteins related to nitrogenase activity in rhizobia were upregulated, probably because the rhizobia sensed the stress and slowed the nitrogen-mediated inhibition of nitrogenase by upregulating key nitrogenase proteins (A0A1L6C021, S6BN46, G7DEZ2) (Appendix A). Bacanamwo and Harper [52] suggested that nitrogen fixation activity in soybean plants is related to the content of asparagine and its metabolites (aspartic acid and glutamic acid). One day after exogenous nitrogen application, ARA decreased, asparagine in the aboveground portions increased, and aspartic acid and glutamic acid in the root nodules increased. In the present study, asparagine in the root nodules increased, and asparagine synthetase (A0A1L3FB30, A0A0A3XKG3) was upregulated during nitrogen treatment (Appendix A). However, ureide decreased, indicating that ureide synthesis was inhibited; this inhibition may have been closely associated with an increase in asparagine. In summary, exogenous nitrogen promoted the synthesis of signaling molecules, including putrescine, NO, and asparagine in root nodules (Figure 9). Further studies are needed to evaluate the sequence of events following increases in nitrogen concentration and the relationship with the regulation of nitrogen fixation in root nodules.

## 4. Materials and Methods

### 4.1. Plant Materials, Growth Conditions, and Treatments

The experiment was conducted at the Northeast Agricultural University experimental station (Harbin, Heilongjiang Province, China, 126°43′ E, 45°44′ N). The seeds were nodulated soybeans (HeiNong40 *Glycine max* L. cv.) and non-nodulated soybeans (WDD01795, L8-4858 *Glycine max* L. cv. obtained from the Academy of Agricultural Sciences in China, Beijing). The experimental nitrogen source was NH_4_NO_3_. We prepared a soybean dual-root system with a unilateral nodulated side. A detailed description of the grafting method and nutrient solution preparation method is provided in Appendix A.

Before the V3 stage, nutrient solution with a nitrogen concentration of 25 mg/L (NH_4_NO_3_ concentration of 71.4 mg/L) was added to both sides of soybean seedlings for culture. From the V3 stage to the V4 stage (for 5 days), the seedlings were divided into two groups treated with or without nitrogen. In the nitrogen-treated group, nutrient solution with a nitrogen concentration of 100 mg/L (NH_4_NO_3_ concentration of 285.6 mg/L) was added to the non-nodulated roots, and nitrogen-free nutrient solution was added to the nodulated root system, which was recorded as NH. Nitrogen-free nutrient solution was added to both sides of the nitrogen-free group, which was recorded as NF.

### 4.2. Sampling Methods

After processing for 5 days, samples were collected. The aboveground portion of the plant along the grafting site was cut, and the underground double-root system was washed with distilled water and then dried with filter paper. Four soybean pots were selected to determine the nitrogenase activity of nodules, and the nodules were removed to determine the number and dry weight. Simultaneously, 13 pots of fresh root nodule samples were frozen in liquid nitrogen after washing with phosphate-buffered solution (PBS) and wrapped in tin foil. Finally, the nodules were transferred to a −80 °C refrigerator and stored. Among them, three pots were used for proteomics analysis, six pots were used for metabolomics analysis, and four pots were used for NO content determination. In addition, four nodules with similar sizes within 6 cm of the grafting site were cut from each treatment to observe the NO distribution in the nodules.

### 4.3. Determination of Acetylene Reduction Assay

Acetylene reduction assay: Soybean root systems were rinsed and dried with filter paper, and all roots and nodules were placed into a 500 mL wide-mouth amber glass bottle. It was then sealed with a rubber plug with a hole, from which 50 mL of air was withdrawn using a syringe, and it was then injected with 50 mL of acetylene (C_2_H_2_, concentration: 99.9%). After 2 h of reaction, the ethylene concentration in the bottle was determined using a gas chromatograph (Model GC7900, Shanghai Techcomp Scientific Instrument Co., Ltd., Shanghai, China). SNA is expressed as μmoles of ethylene formed per gram dry weight of nodules per hour. ARA is expressed as μmoles of ethylene formed per plant per hour.

### 4.4. Production and Content of NO in Nodules

NO production was visualized using 3-Amino, 4-aminomethyl-2′,7′-difluorescein, diacetate (DAF-FM-DA) (Yeasen Biotechnology (Shanghai) Co., Ltd., Shanghai, China). The nodule sections were placed in an appropriate amount of DAF-FM-DA (10 μM) working solution and incubated at room temperature for 1 h in the dark. The excess DAF-FM-DA was removed by washing the nodule sections with PBS. Images were acquired using an inverted fluorescence microscope (Nikon, TS2). An assay kit (Beijing Solarbio Science and Technology) was used to determine NO content.

### 4.5. Proteomics Analysis

Sample preparation: The samples were ground in liquid nitrogen. One milliliter of lysis buffer (7 M urea, 4% sodium dodecyl sulfate (SDS), 1× Protease Inhibitor Cocktail (Roche Ltd., Basel, Switzerland)) was added to the samples, followed by sonication on ice and centrifugation at 13,000 rpm for 10 min at 4 °C. The supernatant was transferred to a fresh tube.

Protein digestion and TMT labeling: The protein concentration of the supernatant was determined using the bicinchoninic acid (BCA) protein assay, and then 100 μg protein per condition was transferred to a new tube and adjusted to a final volume of 100 μL with 100 mM triethylammonium bicarbonate (TEAB). For each sample, proteins were precipitated with ice-cold acetone and then redissolved in 100 μL TEAB.

The proteins were digested with sequence-grade modified trypsin (Promega, Madison, WI, USA), and the resultant peptide mixture was labeled using chemicals from the TMT 6Plex reagent kit. The labeled samples were combined, desalted using a C18 solid-phase extraction (SPE) column (Sep-Pak C18, Waters, Milford, MA, USA) and dried in vacuo.

High pH reversed-phase separation: The peptide mixture was redissolved in buffer A (buffer A: 10 mM ammonium formate in water, pH 10.0, adjusted with ammonium hydroxide) and then fractionated by high-pH separation using an Acquity ultra performance liquid chromatography (UPLC) system (Waters Corporation, Milford, MA, USA) connected to a reversed-phase column. High-pH separation was performed using a linear gradient starting from 0% B to 45% B in 35 min, pH 10.0, adjusted with ammonium hydroxide. Twelve fractions were collected, and each fraction was dried in a vacuum concentrator for the next step.

Low pH nano-HPLC–MS/MS analysis: The fractions were resuspended in 40 μL solvent C (C: water with 0.1% formic acid; D: acetonitrile (ACN) with 0.1% formic acid), separated by nanoLC, and analyzed by online electrospray tandem mass spectrometry. The column flow rate was maintained at 300 nL/min, and the column temperature was maintained at 45 °C. An electrospray voltage of 2.0 kV versus the inlet of the mass spectrometer was used.

The Q-Exactive-HF mass spectrometer was operated in data-dependent mode to switch automatically between MS and MS/MS acquisition. Ten sequential high-energy collisional dissociation (HCD) MS/MS scans with a resolution of 15.0 K were acquired in orbitrap. The intensity threshold was 50,000, and the maximum injection time was 100 ms. The AGC target was set to 100,000, and the isolation window was 1.8 *m*/*z*. Ions with charge states of 2+, 3+, and 4+ were fragmented with a normalized collision energy (NCE) of 30%. In all cases, one microscan was recorded using dynamic exclusion of 30 s. The MS/MS fixed first mass was set at 110.

### 4.6. Metabolomics Analysis

Metabolite extraction: Fifty milligrams of sample was weighed into an Eppendorf (EP) tube, and 1000 μL extract solution (ACN: methanol: water = 2:2:1) with 1 g/mL internal standard was added. After 30 s of vortexing, the samples were homogenized at 35 Hz for 4 min and sonicated for 5 min on ice. The homogenization and sonication cycle was repeated 3 times. Then, the samples were incubated for 1 h at −40 °C and centrifuged at 10,000 rpm for 15 min at 4 °C. The resulting supernatant was transferred to a fresh glass vial for analysis. The quality control (QC) sample was prepared by mixing an equal aliquot of the supernatants from all the samples.

Liquid chromatography–tandem mass spectrometry (LC–MS/MS) analysis: LC–MS/MS analyses were performed using an ultrahigh performance liquid chromatography (UHPLC) system (1290, Agilent Technologies) with an ultra-performance liquid chromatography 1.8 μm High Strength Silica (UPLC HSS T3) column (2.1 × 100 mm, 1.8 μm) coupled to a Q Exactive mass spectrometer (Orbitrap MS, Thermo). In this mode, the acquisition software continuously evaluates the full-scan MS spectrum. The electrospray ionization (ESI) source conditions were set as follows: a sheath gas flow rate of 45 arbitrary unit (Arb), an aux gas flow rate of 15 Arb, a capillary temperature of 400 ℃, a full MS resolution of 70,000, an MS/MS resolution of 17,500, a collision energy of 20/40/60 eV in noncoincidence mode (NCE), and a spray voltage of 4.0 kV (positive) or −3.6 kV (negative).

Data preprocessing and annotation: The raw data were converted to the mzXML format using ProteoWizard and processed with an in-house program developed using R and based on various forms (X) of chromatography coupled to mass spectrometry (XCMS) for peak detection, extraction, alignment, and integration. Then, an in-house MS2 database (BiotreeDB) was applied for metabolite annotation. The cutoff for annotation was set at 0.3.

### 4.7. Statistical Analysis

SPSS 21.0 and Origin 9.0 software were used to analyze the data and generate graphs. All data were tested for normality prior to one-way analysis of variance (ANOVA). Duncan’s multiple range test was used with a significance level of *p* < 0.05. The protein sequences of the selected data were locally searched using Mascot (Matrix Science, London, UK; version 2.3). Mascot was set up to search the UniProt database (Taxonomy: *Bradyrhizobium japonicum*; Glycine max (soybean) (*Glycine hispida*)). Mainly Bradyrhizobium populations reside in the soil environment of Northeast China, and the dominant species is *Bradyrhizobium japonicum* [53,54]. Therefore, the *Bradyrhizobium japonicum* database was searched for rhizobia. DEPs between the treatment groups, as well as the numbers of upregulated and downregulated proteins, were determined by using a FC greater than 1.2 for protein upregulation or smaller than 0.83 for protein downregulation and a *p* value < 0.05 with Student’s *t* test as criteria. DEMs were identified using a variable influence in the projection (VIP) value for orthogonal projections on latent structure-discriminant analysis (OPLS-DA) > 1 and a *p* value < 0.05 as criteria. GO terms were mapped, and sequences were annotated using the software program Blast2GO. Pathway analysis of the proteins and metabolites of interest was performed using the KEGG database (http://www.kegg.jp/ (accessd on 20 November 2021)) and KEGG Automatic Annotation Server (KAAS) software. Based on the information from the KEGG database, the DEPs and DEMs were mapped to KEGG pathways, and then the KEGG annotation and enrichment data obtained from both proteomic and metabonomic analyses were integrated by R software (R Version 3.5.1) and pathway profiling.

## 5. Conclusions

A comparison between soybeans treated with and without nitrogen showed that nitrogen regulated the metabolic pathways of starch and sucrose metabolism, organic acid metabolism, sulfur metabolism, nitrogen metabolism, amino acid metabolism, etc. Additionally, exogenous nitrogen promoted the synthesis of signaling molecules, including putrescine, nitric oxide, and asparagine, in root nodules and inhibited the transformation of sucrose to malic acid; consequently, the rhizobia lacked energy for nitrogen fixation. In addition, exogenous nitrogen reduced cell wall synthesis in the root nodules, thus inhibiting root nodule growth and nitrogen fixation.

## Figures and Tables

**Figure 1 ijms-23-02545-f001:**
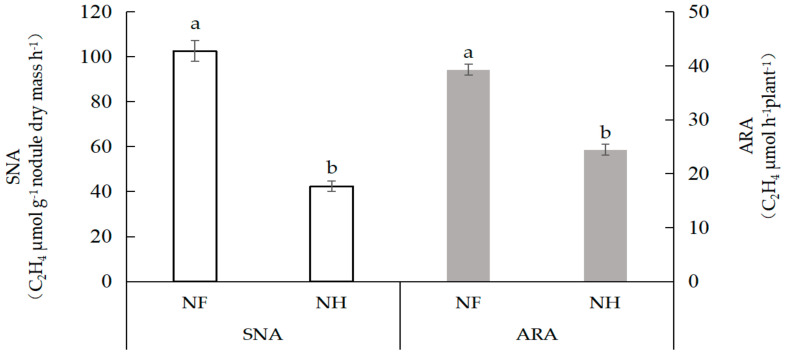
The nitrogenase activity of dual-root soybean nodules. SNA is the specific nitrogenase activity, and ARA is the acetylene reduction activity. Different lowercase letters indicate a significant difference between the treatments at the 5% level.

**Figure 2 ijms-23-02545-f002:**
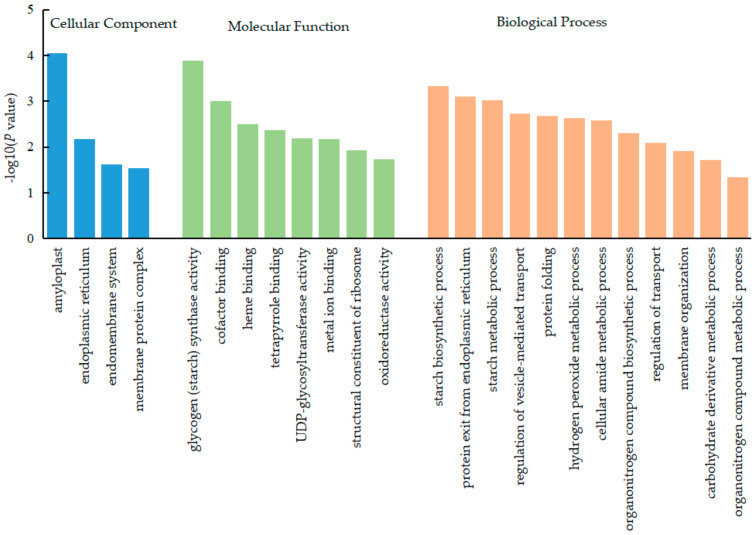
GO annotation statistics of the DEPs in soybean nodules tissue. The GO annotation includes cellular component, molecular function, and biological process, which are color-coded as blue, green, and orange, respectively.

**Figure 3 ijms-23-02545-f003:**
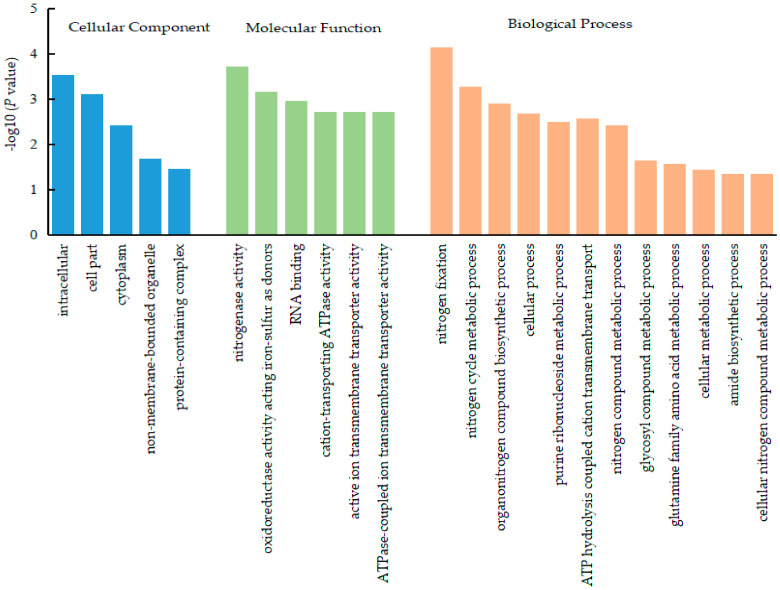
GO annotation statistics of the DEPs in soybean rhizobia. The GO annotation includes cellular component, molecular function, and biological process, which are color-coded as blue, green, and orange, respectively.

**Figure 4 ijms-23-02545-f004:**
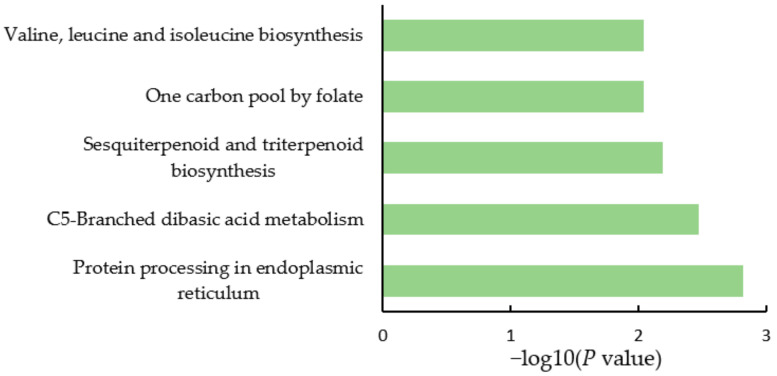
KEGG pathway of the DEPs in soybean nodule tissue after nitrogen supply.

**Figure 5 ijms-23-02545-f005:**
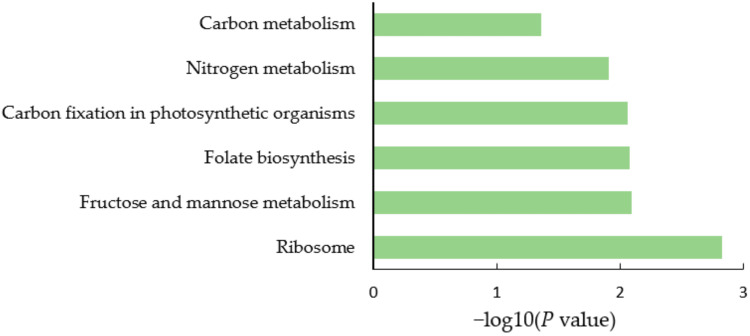
KEGG pathway of the DEPs in soybean rhizobia after nitrogen supply.

**Figure 6 ijms-23-02545-f006:**
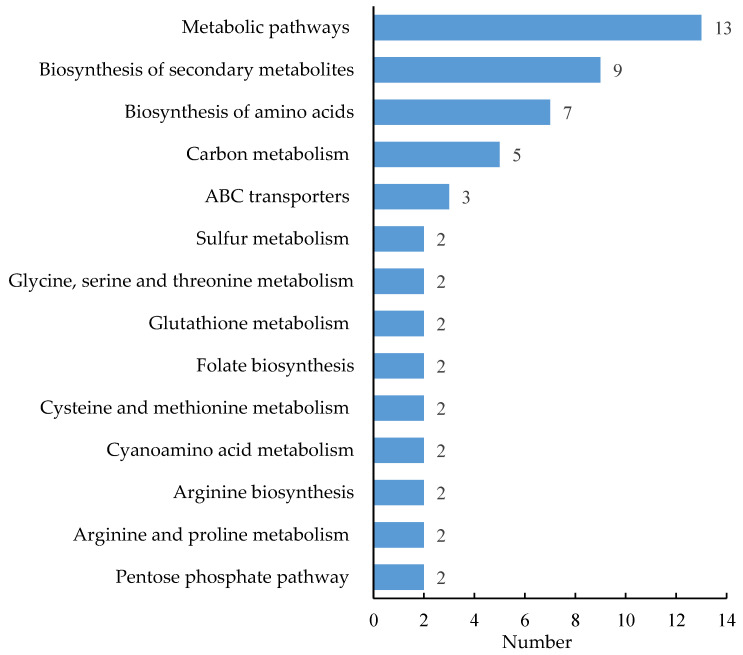
KEGG pathways enriched with DEMs after nitrogen supply.

**Figure 7 ijms-23-02545-f007:**
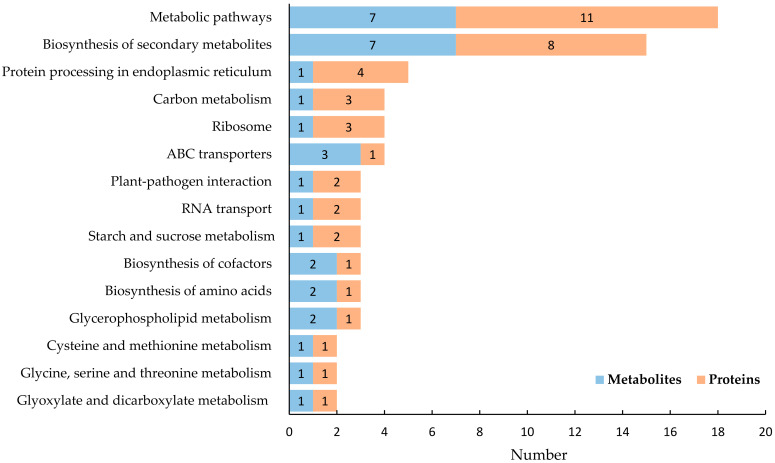
The KEGG pathways involving both the DEPs and DEMs in soybean nodules tissue. Orange indicates the number of DEPs involved, and blue indicates the number of DEMs.

**Figure 8 ijms-23-02545-f008:**
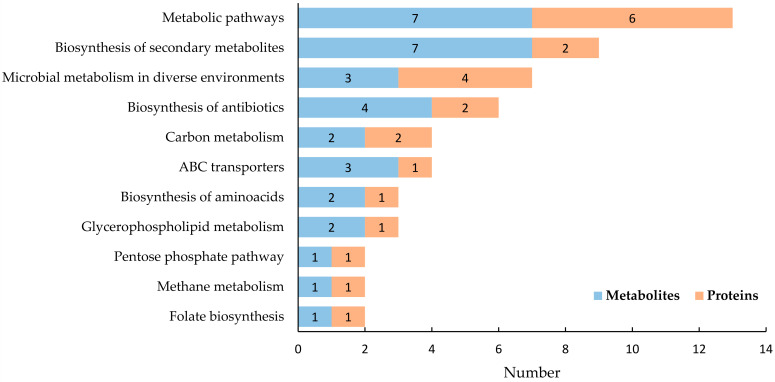
The KEGG pathways involving both the DEPs and DEMs in soybean rhizobia. Orange indicates the number of DEPs involved, and blue indicates the number of DEMs.

**Figure 9 ijms-23-02545-f009:**
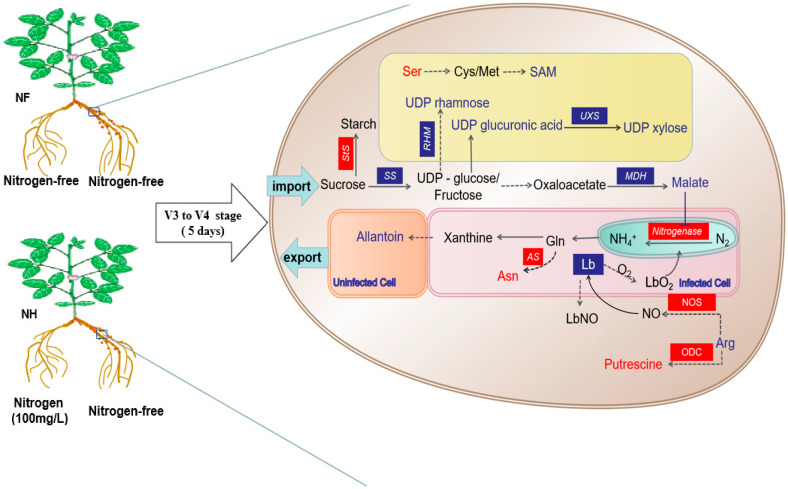
DEPs and DEMs associated with the pathway of nitrogen-regulated pathway in nodules. StS: starch synthase, SS: sucrose synthase, RHM: rhamnose synthase, UXS: UDP-xylose synthase, MDH: malate dehydrogenase, NOS: NO synthase, ODC: ornithine decarboxylase, Lb: leghemoglobin, LbNO: nitrosylleghemoglobin, LbO**_2_**: oxyleghemoglobin, AS: asparagine synthetase, Ser: serine, Cys: cysteine, Met: methionine, SAM: S-adenosylmethionine, Gln: glutamine, Asn: asparagine, Arg: arginine.

**Table 1 ijms-23-02545-t001:** Changes in nodules number and dry weight of dual-root soybeans after nitrogen supply.

Treatments	Nodule Number (Per Plant)	Nodule Weight (g/Plant)
NF	103.0 ± 3.91 a	0.41 ± 0.044 a
NH	98.2 ± 7.71 a	0.37 ± 0.027 a

The data are represented as the mean values ± standard error and independent measurements with four replicates. Different lowercase letters indicate a significant difference between treatments at the 5% level.

**Table 2 ijms-23-02545-t002:** Differential protein identification results of dual-root soybean nodules.

UniProt Database	Upregulated Protein	Downregulated Protein	Total
Glycine max (Soybean) (*Glycine hispida*)	42	10	52
*Bradyrhizobium japonicum*	55	57	112

## Data Availability

The data presented in this study are available in the article and supplementary material.

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
