# Peer review of "Integrated Proteomics and Metabolomics Analysis of Nitrogen System Regulation on Soybean Plant Nodulation and Nitrogen Fixation"

_ijms, 2022, doi:10.3390/ijms23052545_

Round 1

Reviewer 1 Report

Authors applied nitrogen on soybean and performed a proteomic analysis to identify the differences with the control plants which did not receive nitrogen. They identified a wide array of differences.

The study in principle is fine, but the manuscript is not coherent and does not support the conclusions in any way. The result section is completely disconnected from the discussion. They discuss specific genes/ proteins but the results talk about pathways.

Supplementary materials were not included in the version that I received.

The paper is disorganized, with lots of acronyms that have not been described right afterward and some were not described at all. The first line of the result starts with “ Table 1 demonstrates ….” Without explaining what research they did and what was the reason they performed such experiment. The result section is more suitable for a lab meeting presentation where everyone knows about the research. If that was not enough then you get hit with “NF” and “NH” without explaining what they stand for. Took a while to find what these terms mean by searching the whole draft. These should be explained in first order as they appear in the manuscript.

This is also true for DEP. DEM, Gene names, KEGG, HILIC UHPLC-Q-TOF, etc.

Table1 title: is too simplistic, just says what we see in the body of the table.

Line89: what Proteomic technology is that? Why did you use it? and what was the urge of using proteomic technology over other approaches?

Line92: “A total of 164 DEPs…” how did you select these proteins? what was the procedure to select these proteins and what are these?

Line 96: “First”?

Line 112-114: You discussed a wide array of changes belonging to too many pathways. How did you come to such conclusions?

Figures 2-6:

  1. These are simple Histogram, not Abscissa !!
  2. P-value levels are too low. Generally in such studies, a p-value of 3 is considered significant as it gives 99% confidence that differences are significant. Lowering this to “one”, then the probability of “False Positive” increases to a very unacceptable level !
  3. Presumably, each of these bars was constructed using data from several proteins. Then Error bars are necessary to be added to the graphs.
  4. Some labels are incomplete!

Line200: this was not in the results section

Line 217-221: Was not in the result section

Line 229-234: How does lignin relate to nitrogen fixation? If your claim is true, then you should see some phenotypic obvious changes in the nodules. This claim contradicts Table 1.

Line 243-247: How does Fig 9 support this statement? Need good reference and elaborations.

Lines 253-256: a) results of NO concentration were not communicated in the result section. b) How can you make such a conclusion that up-regulation of a protein leads to “actual” NO synthesis?

Author Response

Dear Reviewer:

Thank you for your letter and for the reviewers’ comments concerning our manuscript entitled “Integrated proteomics and metabolomics analysis of nitrogen system regulation on soybean nodulation and nitrogen fixation” (ID ijms-1590802). Those comments are all valuable and very helpful for revising and improving our paper, as well as the important guiding significance to our researches. We have studied comments carefully and have made correction which we hope meet with approval. Revised portion are marked by using "Track Changes" function in Microsoft Word. The main corrections in the paper and the responds to the reviewer’s comments are as flowing:

We have revised the text, figures, tables, references in the manuscript as required, and added the Supplementary Materials to make the manuscript more completed. After careful revision of the article, we chose American Journal Experts for professional English editing, to make the article more in line with the native English expression.

Reviewer 2 Report

Dear Authors,

The research itself is interesting and well presented. The results are beneficial for anyone studying biological nitrogen fixation. Some minor oversights need to be addressed:

  • abstract is more than 200 words,
  • there are some typographical errors such as spaces omitted before references in brackets etc.,
  • acronyms/abbreviations are not defined the first time they appear in the abstract, main text, figure or table captions (see instructions for authors),
  • the data presented in figures does not have to be listed in the text, a reference to the figure is sufficient,
  • GO annotations are not written in full (Figure 2,3,5)
  • Figure 1 – right bracket in the ordinate axis title stands alone,
  • Figure 2 – the ordinate axis title overlaps with numbers,
  • plant materials, growth conditions and treatments are insufficiently described, but I did not get the supplementary materials the text references to,
  • sampling methods are insufficiently described, ‘some samples’ is not a reproducible number,
  • the references are mostly outdated; some more recent studies should be included.

Best of luck!

Author Response

(The authors gave the same response as above.)

Round 2

Reviewer 1 Report

This version was greatly improved compared to the first version and the authors adequately responded to the majority of the issues.

  1. Description pages for Table S1 and S2 need to be translated to English
  2. Table S3 needs an info page to describe the table and terms.
  3. Table S3, “MS1 names” and “MS1 ppm” are blank columns, why are these included in the table?
  4. Respond to point 9 – I know the procedure but my point was among all the pathways came up significant in this study, why did the authors prioritise some proteins over the others?
  5. In response to point 11 authors exactly repeated my point but from the opposite way. Controlling FDR for lower than 1% means having a significant rate of 99% or greater and that is a "–log(p-value) = 3". A p-value that is lower than 3 means FDR greater than 1%. Given the inherent nature of variance for such experiments, this is pushing the boundary for significance too far. We could leave the judgments to the readers.

Author Response

Response to Reviewer 1 Comment

Point 1: Description pages for Table S1 and S2 need to be translated to English

Response 1: Thanks for the reviewer's suggestion, we modified the Table S1 and S2.

Point 2: Table S3 needs an info page to describe the table and terms.

Response 2: Thanks for the reviewer's suggestion, we modified the Table S3.

Point 3: Table S3, “MS1 names” and “MS1 ppm” are blank columns, why are these included in the table?

Response 3: According to the Reviewer’s comments, We removed blank columns MS1 names” and “MS1 ppm”in Table S3.

Point 4: Respond to point 9 – I know the procedure but my point was among all the pathways came up significant in this study, why did the authors prioritise some proteins over the others?

Response 4: Thanks again for the reviewer's question, by reviewing previous research and academic literature, we screened the proteins related to root nodule nitrogen fixation from the identified differential proteins for further in-depth analysis.

Point 5: In response to point 11 authors exactly repeated my point but from the opposite way. Controlling FDR for lower than 1% means having a significant rate of 99% or greater and that is a "–log(p-value) = 3". A p-value that is lower than 3 means FDR greater than 1%. Given the inherent nature of variance for such experiments, this is pushing the boundary for significance too far. We could leave the judgments to the readers.

Response 5: Once again, thanks to the reviewers for their comments, we refer to the P values defined by other scholars in proteomics and metabolomics, which have been listed in the previous response comments. This study is consistent with the relevant literature. We hope that the screening condition of P<0.05 in this manuscript can be retained.